# Application of a Portable Colorimeter for Reading a Radiochromic Film for On-Site Dosimetry

Hiroshi Yasuda [1,*] and Hikaru Yoshida [2]

1 Department of Radiation Biophysics, Research Institute for Radiation Biology and Medicine, Hiroshima University, 1-2-3 Kasumi, Minami-ku, Hiroshima 734-8553, Japan

2 School of Medicine, Hiroshima University, 1-2-3 Kasumi, Minami-ku, Hiroshima 734-8553, Japan

* Correspondence: hyasuda@hiroshima-u.ac.jp

**Featured Application: The present study has demonstrated that simple, accurate on-site dosimetry with a radiochromic film (Gafchromic EBT-XD) can be achieved by using a portable colorimeter (nix pro 2). This finding will extend the potential of radiochromic materials in their applications to various occasions associated with high-dose radiation exposure.**

**Abstract:** Radiochromic films have widely been used for quality assurance (QA) in radiation therapy and have many advantageous features such as self-developing visible coloration, wide dose range and easiness to handle. These features have a good potential for application to other fields associated with high-dose radiation exposure, e.g., verification of various radiation sources used in industry and research, occupational radiation monitoring as a preparedness for radiological emergencies. One of the issues in such applications is the elaborate process of acquisition and analyses of the color image using a flatbed scanner and image processing software, which is desirably to be improved for achieving a practical on-site dosimetry. In the present study, a simple method for reading a radiochromic film by using a portable colorimeter (nix pro 2; abbreviated here "Nix") was proposed and its feasibility for diagnostic X-rays was tested with a commercial radiochromic film (Gafchromic EBT-XD). It was found that the color intensities of red and green components of EBT-XD were successfully measured by Nix over a wide dose range up to 40 Gy. Though some angle dependence was observed, this error could be well averted by careful attention to the film direction in a reading process. According to these findings, it is expected that the proposed on-site dosimetry method of combining a radiochromic film and a portable colorimeter will be practically utilized in various occasions.

**Keywords:** radiochromic; gafchromic; film; EBT-XD; portable colorimeter; nix; X-rays

## 1. Introduction

Radiochromic films undergo a visible color change directly after exposure to radiation without chemical processing for the development of the image by a dye creation through a radiation-triggered polymerization process. This feature has made radiochromic films a preferable tool for two-dimensional dose verification and quality assurance (QA) of external beams such as photons, electrons and protons used in radiation therapy [1–30].

A commercial radiochromic film (Gafchromic, International Specialty Products, Wayne, NJ, USA) appeared for industrial use in the late 1980s [31]. While this polydiacetylene-based product had a good dose-response, the initial products of Gafchromic film had some issues related to sensitivity, uniformity and reproducibility in color development [32–34]. These issues have been well solved by continuous improvements of dosimetric properties in subsequently developed models (HD-810, MD-55, HS and EBT series) [20,22,25,34–36] by gaining higher spatial resolution, nearer tissue equivalence and weaker energy dependence for different quality radiations. The recent EBT model is relatively insensitive to visible light, which has offered ease of handling under normal indoor light conditions. In addition, its reasonable cost is a considerable advantage for practical use [35,36].

These preferable features of radiochromic films have good potential for application to many other situations other than radiation therapy. The self-developing coloration soon after exposure to any type of radiation would be effectively used for periodical dose verifications of various radiation sources in industry and research. It is also expected that robust, light-insensitive radiochromic films would be desirably used for individual monitoring of occupational radiation exposure of workers who routinely handle high-level radiation sources or those who will be involved in the initial responses to nuclear/radiological emergencies. In these occasions, the ability of a sensitive radiochromic film that can detect in real time a high-dose exposure according to the color change recognized by the naked eye would be helpful for earlier recognition of the occurrence of a radiological accident, which enables us to save the lives of highly exposed casualties by taking necessary medical actions without delays.

In the prompt decision on the best medical treatment in such an urgent situation, it is indispensable to know immediately the accurate radiation dose and exposure timings of the individual victim. Though there are several dosimetry techniques applicable to such situations [37–40], the dose assessments with these techniques are performed in retroactive manners and have limitations for on-site use as the measurements need large, non-portable equipment to read the thermally/optically stimulated luminescence or electron spin/paramagnetic resonance signals from the samples exposed to radiation. Some approaches using biological samples such as tooth enamel and peripheral blood lymphocyte need several days for processing the samples including the elaborate dose calibration. Although real-time monitoring would be possible if the subject carried an electronic personal dosimeter, such small electronic devices cannot accurately respond to pulsed high-dose radiation exposure because of the count-loss problem, which could be averted by using a radiochromic film applicable to a wide dose range.

For using radiochromic films for the on-site dosimetry required under such emergency situations, however, the current procedure of measuring films is an issue in terms of practicality, as it requires elaborate tasks for the acquisition of image data using a flatbed scanner and subsequent analyses on color intensities using an image processing software [20,25,36]. It would be beneficial if we could measure the color intensities of any radiographic films more simply and shortly with portable devices only. It is expected that such a simple dosimetry technique would work also as a supplementary tool for the verification of delivering doses of external radiotherapy beams prior to detailed measurements of two- or three-dimensional dose distributions. With these thoughts, we propose here a simple method using a portable colorimeter for reading a radiochromic film on-site immediately after radiation exposure and investigate the feasibility of the proposed method through experiments with diagnostic X-rays.

## 2. Materials and Methods

### 2.1. Materials for Dosimetry

A commercially available radiochromic film (Gafchromic EBT-XD, ISP, Wayne, NJ, USA) (Figure 1) was employed in the present study. EBT-XD is a Gafchromic film released in 2015 as a recent product of the EBT series model (EBT, EBT2, EBT3 and EBT-XD) which appeared in 2004. EBT-XD has some better features than the prior product Gafchromic EBT-3 [18,36] in view of delivering beam dosimetry in radiation therapy. For example, the optimal dose range of EBT-XD is 0.4 to 40 Gy and its dynamic dose range is 0.1 to 60 Gy; these ranges are higher than those of EBT-3 (0.1 to 10 Gy and 0.1 to 20 Gy, respectively), while the physical structures of both products are nearly the same. The EBT-XD film have a 25 μm active layer sandwiched with two 125 μm-thick polyester substrates. The size of its active particle is 2 to 4 μm in length with a diameter of 1 to 2 μm; such small-size particles are considered to work for reducing light scattering and polarization, leading to fewer lateral response artifacts, the so-called 'orientation effect' which is characterized by a change in the measured color intensity of a scanned film depending on its orientation on the scanner [21,25,41–44].

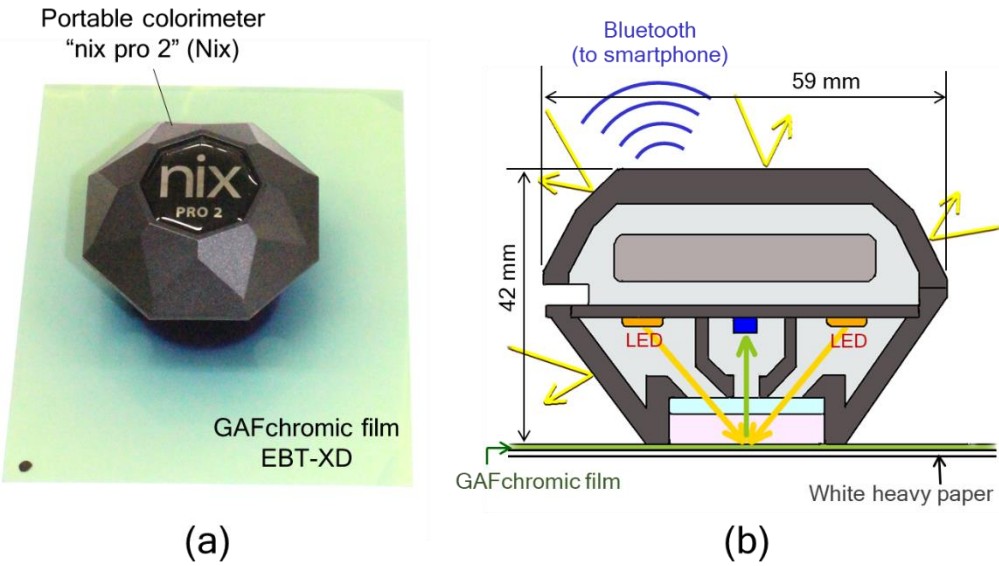

**Figure 1.** The materials used for this study (**a**): Gafchromic film "EBT-XD" and a portable colorimeter "nix pro 2" (abbreviated to "Nix" here) and a cross-section drawing of Nix (**b**). The black frame structure of Nix blocks well the ambient light; two LED sources (orange part) cast light beams and a color sensor (blue part) detects the reflected light; the data of color intensities measured as RBE and CMYK components are transferred to a smartphone via Bluetooth.

The color intensities of EBT-XD were measured with a commercially available portable colorimeter ("nix pro 2", Nix Sensor Ltd., Ontario, Canada; abbreviated to "Nix" hereafter) shown in Figure 1a. While a flatbed scanner has generally been used for acquiring the two-dimensional images of Gafchromic films in radiation therapy [13,20,23,25,36,45], the authors first employed the portable colorimeter here with an intention of achieving simple on-site dosimetry that could be utilized in other fields. It should be noted that, since the aperture size of Nix for detecting the reflected light from a sample surface is 14 mm in diameter, the conventional method using a flatbed scanner will be needed for precise two-dimensional dosimetry.

Nix has the shape of a hexagon with dimensions of 60 mm in height × 42 mm in width and a weight of 43 g. It includes a rechargeable lithium-ion battery and can perform more than 100 scans per single charge. Nix measures the color intensities of any flat material cutting the ambient light (Figure 1b), and the data of color intensity measured as RBE in a few seconds and CMYK components are transferred via Bluetooth to a smartphone having the operating system of Android or iOS and immediately displayed on its screen. These data can be saved into the smartphone and exported for further analyses.

The features of Nix such as the small size, light weight, easiness to handle and relatively low price (350 USD) are preferable for practical application to on-site dosimetry. Particularly, the good portability is quite beneficial for the aim of the present study, as anyone can easily carry this device in their pocket to prepare for a possible occasion of a radiological incident. The high performance of Nix in reading color intensities has been well validated in previous studies for the determination of soil compositions [46–49] and evaluation of beef freshness [50,51].

### 2.2. Irradiation and Film Analyses

The EBT-XD films were irradiated with 160 kV X-rays generated in a commercial X-ray irradiator ("CP-160", Faxitron, Wheeling, IL, USA) at 10 levels of absorbed doses: 1, 2, 4, 7, 10, 15, 20, 25, 30 and 40 Gy. The dose rate of the static X-ray beam was adjusted by changing the distance between the source and the film so that the irradiation time would be more than 2 min in every case. The field diameter was more than 5 cm even at the shortest distance. The color intensities of the irradiated EBT-XD films placed on a white heavy

paper were measured with Nix within a few hours and the data were recorded by using an exclusive application on the smartphone as RGB color components: RGB, red, green and blue. In reading the films, a thick white paper for drawing (Kent paper) was placed under the EBT-XD film (Figure 2a).

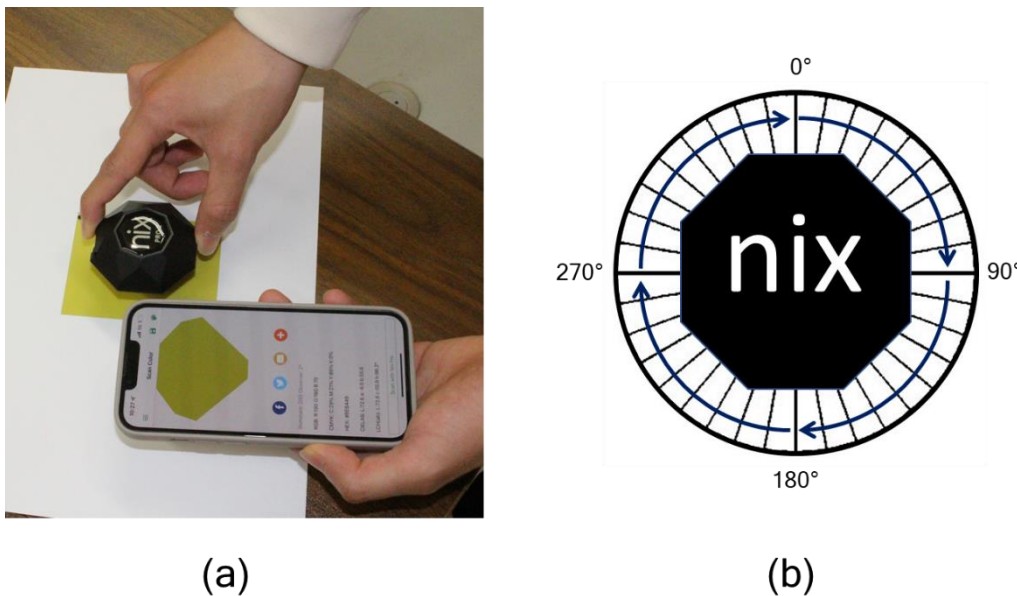

(a)                                                      (b)

**Figure 2.** A scene of reading the EBT-XD film using Nix (**a**) with a smartphone application for reading exclusively the data from Nix via Bluetooth. The measurements of the films in a few hours after irradiation with 160 kV X-rays. Angle dependence of measured color intensities was examined by rotation of Nix with 15° intervals in a clockwise direction (**b**).

There are artifacts associated with the color digitization of Gafchromic films scanned with flatbed scanners, which are known as 'orientation effect' and 'lateral response artifact' [1,21,25,36,41–44]. This orientation effect is characterized by a change in response as a function of the orientation of the film on the scanner bed. Thus, the dependence of color intensity measurements on the angle in measurement was also examined for Nix by rotating with 15° intervals in a clockwise direction (Figure 2b).

Since the radiochromic films are measured by using a flatbed document scanner [20,25,36], the irradiated films were scanned also with a commercial flatbed scanner (EPSON GT-X900, Seiko Epson Corp., Nagano, Japan) at the same timings as the Nix readings. A 48-bit color image of each film was read in reflection mode with a resolution of 600 dpi using built-in scanner software for Windows. The same-type scanners of the EPSON GT series have been used for scanning Gafchromic films in some preceding studies [13,52,53]. For making accurate comparisons of the color intensities measured with Nix, the EBT-XD films were covered with the same white heavy paper while being scanned with the flatbed scanner.

Statistical analyses of the measured data and productions of plot graphs including regression curves were made using commercial data analysis software (IGOR Pro, Wave-Metrics, Inc., Portland, OR, USA).

## 3. Results and Discussions

### 3.1. Selection of Color Components

It should be noted that the color intensity of a radiochromic film has generally been evaluated with the optical density change (*netOD*) after radiation exposure which is calculated as follows [23,36]:

$$netOD(D) = -\log_{10} \frac{PV(D)}{PV(0)} \tag{1}$$

where *PV(D)* represents the pixel's color channel value of the film exposed to radiation at dose *D*; thus *PV(0)* means that of the unirradiated film which is measured before irradiation. The pixel color value is normally obtained for each of the RGB channels (red, green and blue components).

The authors judged that this way of calculation using the logarithmic functions Equation (1) was too complex in view of practicality and would not fit the aim of the present study, i.e., the development of a simple on-site dosimetry method. Thus, the change in color intensity (Δ*I*) caused by radiation exposure was simply calculated here from the readings obtained with the portable colorimeter as follows:

$$\Delta I = I(0) - I(D) \tag{2}$$

where *I(D)* is the measured intensity of the respective color channel at *D*. More concretely, *I*(0) was given as the value measured by Nix before irradiation and *I*(*D*) was given as that a few hours after irradiation; it was confirmed that the levels of radiation-induced colorations of EBT-XD had been stable for 24 h after irradiation in all cases. We should note that *netOD* would preferably be used in place of Δ*I* if smartphone software which could automatically calculate the netOD value in a simple operation process was available.

Figure 3 shows the dose responses of the color intensities (RGB, red, green and blue) obtained by Nix with the EBT-XD film irradiated with up to 40 Gy of 160 kV X-rays. The regression curves indicated there were obtained with the following function:

$$\Delta I = a \times \left(1 - e^{(-b \cdot D)}\right) c \tag{3}$$

where *a*, *b* and *c* are the fitting parameters that are empirically determined through regression analyses. As seen in the figure, the red and green components showed good dose responses with small error bars; values of the coefficient of variation (CV) defined as the ratio of the standard deviation to the mean value were less than 3% in all cases. On the other hand, the blue component showed a unique, flat dose response with larger error bars, which indicates a lower sensitivity of this component in radiochromic reactions. As a mixture of the responses of red, green and blue components, the sensitivity of RGB was in the middle of those of these three components.

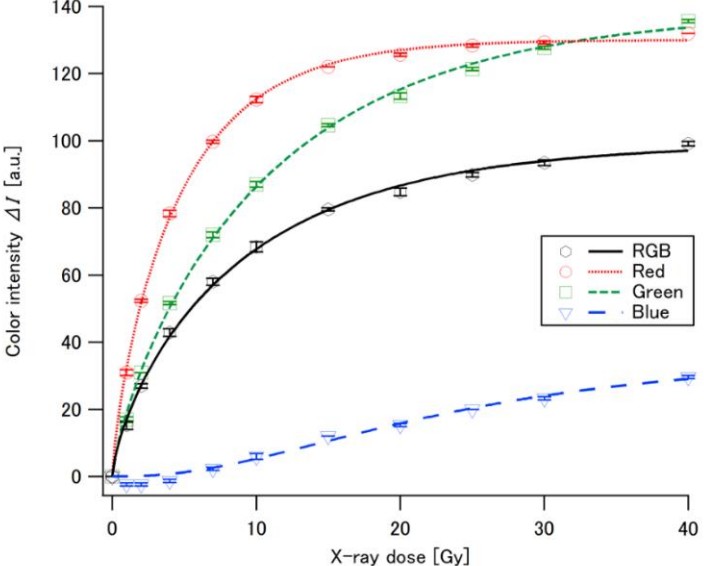

**Figure 3.** Dose dependences of the color intensities of EBT-XD irradiated with 160 kV X-rays at up to 40 Gy as to RGB, red, green and blue components measured with Nix at the angle of 0°.

While the dose responses of both the red and green components showed convex upward curves, not linear ones, it was considered that a unique radiation dose could be determined directly from each of the measured color intensities as they were monotonically increasing with dose. In a comparison of the red and green components, the red component increased more sharply and reached a saturated status around 10 Gy, while the green component showed a constantly increasing curve up to 40 Gy or higher. This difference qualitatively agrees with the findings in previous studies [21,42] and thus it was judged that these two components would be appropriately used for dose assessment in this dose range (~40 Gy) in case the color intensities of the films be measured by Nix.

### 3.2. Angle Dependence and Dose Response

The color intensities of red and green components measured with Nix being rotated with 15° intervals are shown in Figure 4 for four dose levels (0, 4, 10 and 40 Gy) of 160 kV X-rays. In the calculation of the color intensity ($\Delta I$) at each angle, the intensity of the unirradiated film ($I(0)$ in Equation (2)) was given as that measured at 0°. As seen in the figure, systematic angle dependence was clearly observed at all dose levels in both color components. The absolute variation ranges were similar in every case regardless of dose levels, which indicates this angle dependence is attributable to the anisotropic light scattering on the surface of the Gafchromic film. Accordingly, the relative errors became higher at lower doses according to the reduced color intensities. The angle dependence was seen in unirradiated (pre-irradiation) samples also. These findings indicate that the so-called 'orientation effects' or 'lateral response artifacts' reported for the Gafchromic EBT films measured with flatbed scanners [1,21,25,36,41–44] also appear in measurements with a portable colorimeter.

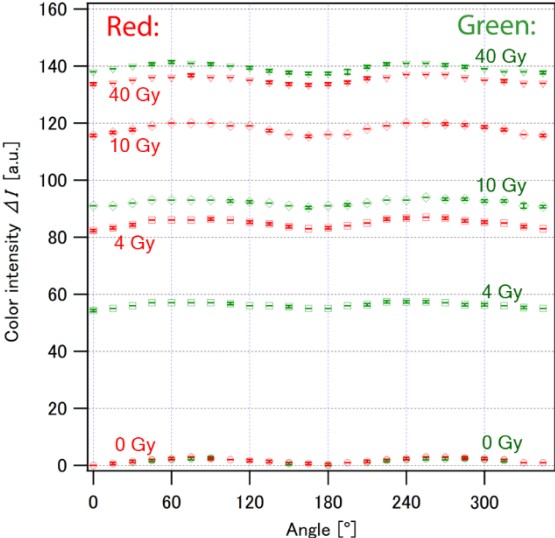

**Figure 4.** Angle dependences of the color intensities as to the red and green components of EBT-XD measured with Nix with 15° intervals after irradiation with 160 kV X-rays at 0 (control), 4, 10 and 40 Gy.

To confirm the effects of the angle dependence on dose assessments, the dose responses of red and green components were compared among the data obtained at four different angles (0°, 90°, 180° and 270°) as shown in Figure 5. From these plots, it was considered that some errors up to several percent could be potentially caused by the uncontrolled orientation of Nix in the measurement of EBT-XD. It is expected, however, that this error attributed to the angle dependence can be averted by the careful handling of Nix paying and attention to the film orientation during reading. More concretely, a user can obtain the color intensity for a dosimetry purpose by keeping the same orientation of the Gafchromic film when reading it before and after irradiation.

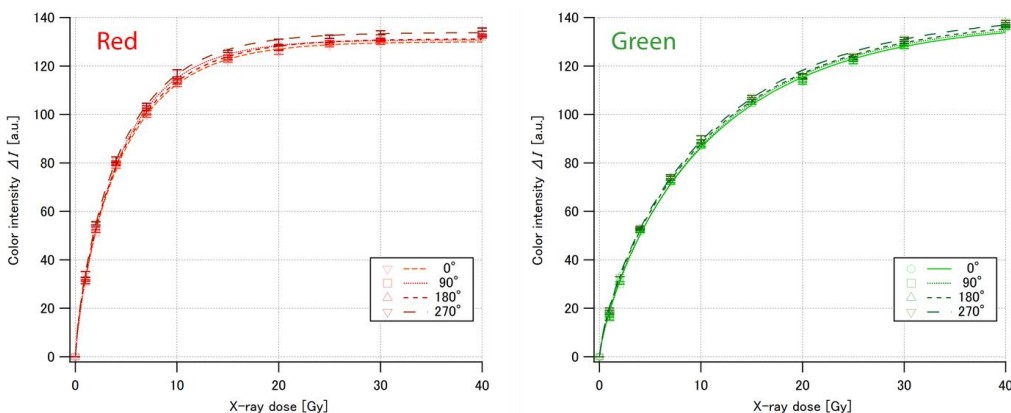

**Figure 5.** Dose dependence of the color intensities (**left**): red component, (**right**): green component) of EBT-XD irradiated with 160 kV X-rays at up to 40 Gy and measured with Nix at four angles: 0°, 90°, 180° and 270°.

*3.3. Comparison with a Flatbed Scanner*

In Figure 6, the red and green color intensities of X-ray irradiated EBT-XD films measured with a conventional flatbed scanner (EPSON GT-X900) are compared to those measured with Nix as to two orientations: 0° (above) and 90° (below). The color intensities measured with the flatbed scanner were constantly higher than those with Nix, which implies that the reflection patterns of the LED light used for image scanning were different between the two devices. From these results, the performance of Nix in reading the Gafchromic films was judged to be comparable to that of the conventional flatbed scanner.

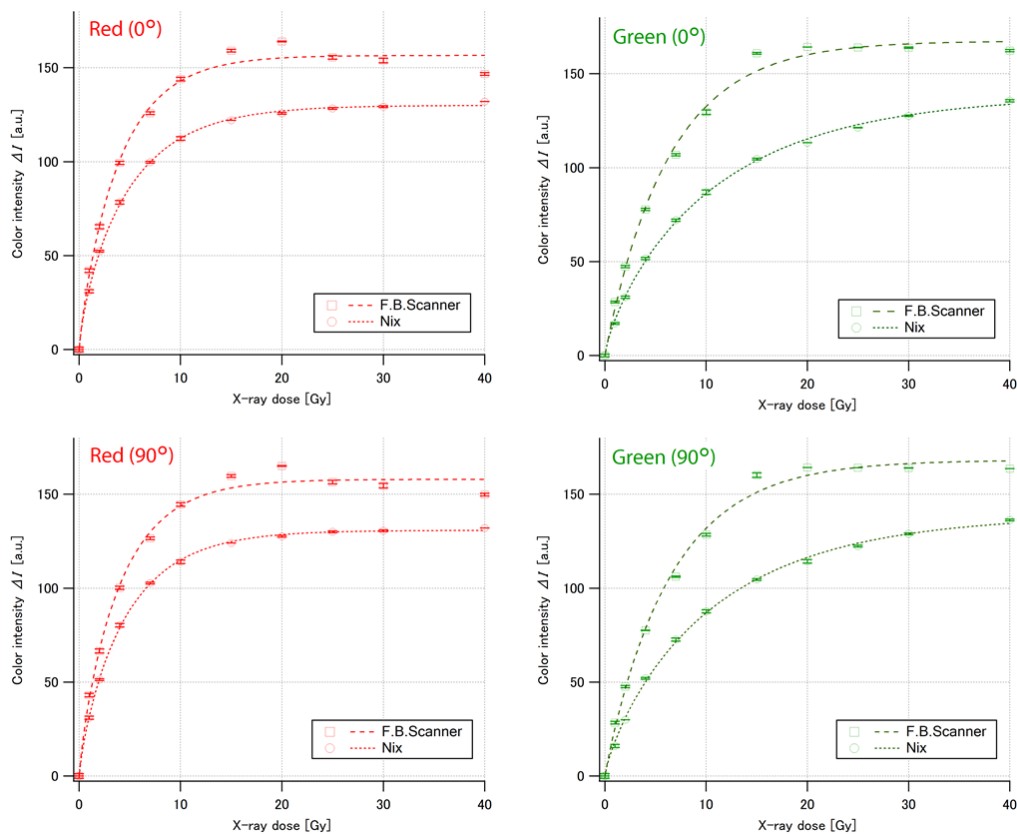

**Figure 6.** Comparison between the dose dependencies of the color intensities (**left**): red component, (**right**): green component) of EBT-XD measured with Nix and those with the flatbed scanner at the angles of 0° (**above**) and 90° (**below**).

It was found that both red and green components obtained with the flatbed scanner tended to saturate at unexpectedly lower dose levels, i.e., around 10 Gy and 15 Gy, respectively. This tendency (saturation at lower dose) observed in the flatbed scanner data can partly be attributed to the selection of the reflection mode, not the transmission mode, as it was reported that the color intensities of the EBT-3 Gafchromic film scanned with a similar-type flatbed scanner (EPSON 10000XL) in reflection mode had more limited dose range and slightly larger uncertainty than those obtained in transmission mode, while the reflection mode scanning showed higher sensitivity and could lead to more widespread use for the purpose of radiochromic film dosimetry [54]. These findings indicate that the proposed dosimetry method using Nix for reading a radiochromic film (EBT-XD) is promising for on-site dosimetry, i.e., the dosimetry that can be achieved with portable items only. It is expected that this approach would be effectively applied to other radiochromic materials that would be flexibly selected depending on the conditions of real or potential radiation exposures, such as possible dose range, radiation species, mainly affected body parts, etc.

## 4. Conclusions

In the present study, the authors proposed a simple method for on-site dosimetry using a radiochromic film (Gafchromic EBT-XD) coupled with a portable colorimeter (Nix) for reading and investigated its performance and practicality for up to 40 Gy of 160 kV X-rays. It was confirmed that the color intensities of red and green components simply measured with Nix had favorable dosimetric properties such as a monotonical increase with dose, good reproducibility and low variability (CV < 3%) in addition to excellent portability. While it was found that the readings with Nix show some angle dependence such as the orientation effects reported in previous studies using flatbed scanners, this issue could be well averted by careful handling of a radiochromic film.

These findings indicate the good potential of the proposed on-site dosimetry method for application to various occasions of real or potential high-dose radiation exposure. It is expected that the feature of simple and rapid dose quantification would be desirably used for QAs of radiation sources used in industry and research as well as medicine. The radiation-induced color change that can be detected by the naked eye would enable us to take immediately proper medical actions for a patient who received a high-dose radiation exposure in radiological emergencies. Moreover, such a simple dosimetry technique is expected to be utilized as a supplementary tool in QA of the external radiotherapy beams that are delivered repeatedly to many cancer patients with short-time intervals. Further studies with different exposure situations and different quality radiations are needed to clarify the effectiveness and limitations of this method in those possible applications.

**Author Contributions:** Conceptualization, H.Y. (Hiroshi Yasuda); material preparation and maintenance, H.Y. (Hiroshi Yasuda) and H.Y. (Hikaru Yoshida); irradiations and measurements, H.Y. (Hiroshi Yasuda) and H.Y. (Hikaru Yoshida); data analysis, H.Y. (Hiroshi Yasuda) and H.Y. (Hikaru Yoshida); writing—original draft preparation, H.Y. (Hiroshi Yasuda); writing—review and editing, H.Y. (Hikaru Yoshida); funding acquisition, H.Y. (Hiroshi Yasuda). All authors have read and agreed to the published version of the manuscript.

**Funding:** This research was supported by JSPS KAKENHI Grant Number 18KK0147 and the Program of the Network-type joint Usage/Research Center for Radiation Disaster Medical Science funded by the Ministry of Education, Culture, Sports, Science, and Technology (MEXT) of Japan and Hiroshima University.

**Institutional Review Board Statement:** Not applicable.

**Informed Consent Statement:** Not applicable.

**Data Availability Statement:** Not applicable.

**Conflicts of Interest:** The authors declare no conflict of interest.

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
