# Peer review of "Application of a Portable Colorimeter for Reading a Radiochromic Film for On-Site Dosimetry"

_applsci, doi:10.3390/app13084761_

Round 1
Reviewer 1 Report
The topic of radiation therapy is very hot and significant. As a result, developing techniques to detect radiation dose is also important. In the present study, the authors proposed a simple method for on-site dosimetry using a radiochromic film (Gafchromic EBT-XD) coupled with a portable colorimeter (Nix) for reading and investigated its performance and practicality for up to 40 Gy of 160 kV X-rays. The work is done well and all data are reliable. The referee suppots its publication after considering the following issues.
1 The authors investigated the dose dependences of the color intensities. Is there a linear relationship between the dose and signal which could be used to directly determine the radiation dose.
2 As a sensor technology, the response time should be given.
3 The selectivity towards the radiation (such as wavelength) should be considered.
Reviewer 2 Report
General considerations
The aim of this work regards an application of a portable colorimeter for reading a radio-chromic film for on-site dosimetry.
Radio-chromic films are nowadays of interest for dosimetry and this work is potentially interesting as a simple method to have fast results.
Remarks:
1. page 3 “The EBT-XD films were irradiated ... at 10 levels of absorbed doses: 1, 2, 4, 7, 10, 15, 20, 25, 30 and 40 Gy..” Please, add a short description of the method used to calibrate the X-ray generator.
2. page 5 “cases. I should note that ...”. I?, please change in we.
3. page 6, Figure4. I suggest to put red and green curves on the same chart in order to better highlight the similarity between the two.
4. page 7 “In Figure 6, the red and green color intensities of X-ray irradiated EBT-XD films measured with a conventional flatbed scanner (EPSON GT-X900) are compared....”. Because not all scanners are suitable in film dosimetry, please, add in the paper the reasons of this choice and the property of this scanner at least in terms of: bits, spatial resolution, and software used.
Reviewer 3 Report
The manuscript "Application of a Portable Colorimeter for Reading a Radiochromic Film for On-Site Dosimetry" by Hiroshi Yasuda and Hikaru Yoshida proposes a new method for evaluating the dose exposure of X-ray radiation up to 160 keV using a portable colorimeter Nix pro 2 to read a radiochromic EBT-XD film. The experiments were conducted over a wide dose range of up to 40 Gy, and the authors presented the dose dependence of the color intensity for the red, green, and blue components measured by the Nix portable colorimeter. The authors evaluated the effect of the rotation angle of the Nix colorimeter on dose assessments and demonstrated a good monotonic increase with dose at four different angles of 0°, 90°, 180°, and 270°. The authors compared the intensity of the red and green components of the EBT-XD gafchromic film measured by the Nix colorimeter with those read by a conventional flatbed scanner and observed a similar trend, but with lower intensity in the case of the Nix portable colorimeter. The study demonstrates that using portable devices for reading gafchromic film is a promising method for on-site dosimetry.
While the work is well-structured and comprehensive, there are concerns regarding the limitations of the Nix colorimeter and the definition of "inappropriate orientation." Specifically, the authors did not mention the spatial resolution of the Nix colorimeter or the minimum exposure area on gafchromic film required for precise measurement. It would be beneficial for the manuscript to emphasize these limitations.
Furthermore, in the "Angle Dependence and Dose Response" section, the authors stated that "some error of several percent could be potentially caused by the inappropriate orientation of Nix in the measurement of EBT-XD." However, the authors did not explain what they considered an "inappropriate orientation."
Overall, I do recommend the publication of the manuscript. I suggest that the authors address the above-mentioned limitations and provide clarification on the definition of "inappropriate orientation" in their manuscript.
